

# Beyond harm's reach? Submersion of river turtle nesting areas and implications for restoration actions after Amazon hydropower development

Darren Norris[1,2,3], Fernanda Michalski[2,3,4] and James P. Gibbs[5]

[1] School of Environmental Sciences, Federal University of Amapá, Macapá, Amapá, Brazil
[2] Ecology and Conservation of Amazonian Vertebrates Research Group, Federal University of Amapá, Macapá, Amapá, Brazil
[3] Postgraduate Programme in Tropical Biodiversity, Federal University of Amapá, Macapá, Amapá, Brazil
[4] Instituto Pró-Carnívoros, Atibaia, São Paulo, Brazil
[5] Department of Forest and Environmental Biology, State University of New York (SUNY), Syracuse, NY, United States of America

Corresponding author
Darren Norris,
darren.norris@unifap.br

## ABSTRACT

The global expansion of energy demands combined with abundant rainfall, large water volumes and high flow in tropical rivers have led to an unprecedented expansion of dam constructions in the Amazon. This expansion generates an urgent need for refined approaches to river management; specifically a move away from decision-making governed by overly generalized guidelines. For the first time we quantify direct impacts of hydropower reservoir establishment on an Amazon fresh water turtle. We conducted surveys along 150 km of rivers upstream of a new dam construction during the low water months that correspond to the nesting season of *Podocnemis unifilis* in the study area. Comparison of nest-areas before (2011, 2015) and after (2016) reservoir filling show that reservoir impacts extend 13% beyond legally defined limits. The submerged nesting areas accounted for a total of 3.8 ha of nesting habitat that was inundated as a direct result of the reservoir filling in 2016. Our findings highlight limitations in the development and implementation of existing Brazilian environmental impact assessment process. We also propose potential ways to mitigate the negative impacts of dams on freshwater turtles and the Amazonian freshwater ecosystems they inhabit.

## INTRODUCTION

Freshwater turtles are under threat from the alteration of rivers, habitat loss, climatic changes and anthropogenic changes to the landscape. In aquatic systems, changes in upstream land use and the placement of dams have had significant impacts on ecosystems and turtle populations worldwide (*Castello et al., 2013*; *Lees et al., 2016*; *Rhodin et al., 2011*; *Rödder & Ihlow, 2013*). Despite this, the literature on environmental impact assessments of dams on freshwater turtles is limited, as most dams are constructed before any baseline ecological data were collected (*Keck et al., 2017*; *Poff & Zimmerman, 2010*; *Sousa Júnior et al., 2016*).

There is an urgent need for refined approaches to river management, and a move away from decision-making governed by overly generalized rules of thumb (*Keck et al., 2017*; *Kuehne et al., 2017*; *Latrubesse et al., 2017*). Hydroelectric expansion generates myriad social (*Richter & Thomas, 2007*; *Zedler & Callaway, 1999*), economic and environmental changes (*Freeman, Pringle & Jackson, 2007*; *Kingsford, 2000*; *Molle, 2009*). Generally dams accumulate toxins and release greenhouse gases, which combined with regional deforestation, can generate drastic changes in local and regional climates (*Guimberteau et al., 2017*; *Stickler et al., 2013*). These changes have disproportionately large effects on the communities and biodiversity that are nearest to the installation site. This is particularly true in the Amazon basin, where freshwater ecosystems are vital components and hydrological connectivity with both aquatic and terrestrial ecosystems makes them susceptible to a wider range of anthropogenic impacts at local and regional scale (*Castello et al., 2013*; *Latrubesse et al., 2017*). Thus, the debate as to whether hydropower is renewable continues (*Fearnside, 2016*; *Ferreira et al., 2014*; *Kahn, Freitas & Petrere, 2014*; *Winemiller et al., 2016*).

Although some nations have started to remove dams (e.g., 1,300 dams removed in USA as of 2015) at the global-scale dams and water resources challenges remain all too common in economically developed (e.g., Australia) and developing (e.g., Brazil with its large economy but low per capita GDP) nations (*Sousa Júnior et al., 2016*; *Tundisi et al., 2014*). The 6.15 million km$^2$ Amazon River basin is by far the largest river basin in the world, with the Amazon River discharge contributing more than 15% of the total discharge of all rivers (*FAO, 2016*) playing a key role in maintaining global climate and hydrological cycles. Across the Amazon, transnational river systems and a lack of coordinated management of aquatic systems may result in losses to unspecified levels of biodiversity (*Castello et al., 2013*; *Fearnside, 2016*; *Ferreira et al., 2014*; *Latrubesse et al., 2017*; *Sousa Júnior et al., 2016*; *Winemiller et al., 2016*). Ensuring the social and environmental sustainability of the myriad developments across Amazon waterways is therefore critical for future regional and global well-being (*Kahn, Freitas & Petrere, 2014*; *Latrubesse et al., 2017*; *Tilt, Braun & He, 2009*).

Environmental impact assessment (EIA) systems in Brazil are at a crossroads (*Ferreira et al., 2014*). Environmental impact assessments are the mechanism used to ensure environmental sustainability of major development projects. The purpose of impact assessment is to evaluate whether a stressor has and/or will change the environment, which components are adversely affected, and to estimate the magnitude of the effects. Impact assessments are best based on before-after control-impact (BACI) design (*Smith, 2006*). An appropriately applied BACI design is considered optimal to help isolate the effect of the development from natural variability. Yet due to myriad financial, logistical and political reasons BACI-designed studies are relatively rare and empirical impact assessments are most commonly based on so-called space-for-time substitutions. There is increasing evidence that such space-for-time assessments can systematically underestimate impacts of stressors/changes (*França et al., 2016*; *Gonzalez et al., 2016*).

Effective EIAs of hydropower development require quantification of how far upstream and downstream impacts can reach, as this knowledge is required to directly inform specific strategies and solutions to mitigate the myriad adverse effects on biodiversity and freshwater ecosystems. The spatial extent of upstream impacts depends on the local system

channel geometry, channel slope, and height of the dam. New dams present a unique challenge to freshwater turtles in the region as reservoir formation and flow changes can drastically alter feeding and breeding habits, due to irreversible changes in phenological rhythms of the flooded forest, and submerging of nesting beaches used by Amazonian freshwater turtles (*Castello et al., 2013*; *Lees et al., 2016*).

Our objective was to identify the distance at which the reservoir from a medium-sized dam impacted populations of the yellow spotted river turtle (*Podocnemis unifilis*). We quantified and mapped nesting areas pre- and post-reservoir formation to identify the spatial limit of reservoir flooding on this species. To maximize strength of inference our study employed a before-after comparison, which is lacking in most studies of hydropower development (*Poff & Zimmerman, 2010*). Based on project outcomes, we discuss potential ways to mitigate the negative effects of hydropower development on Amazonian freshwater turtles.

### Ethical statement

Ethical approval was not required for our noninvasive study, as we did not collect any biological sample nor interfere with the behavior of the study species. Permission to collect observational data from river turtle nest-areas was provided by research permit number IBAMA/SISBIO 49632-1 and 49632-2 to DN and FM, issued by the Instituto Chico Mendes de Conservação da Biodiversidade (ICMBio). Interviews with local residents were approved by IBAMA/SISBIO (permits 45034-1, 45034-2, 45034-3) and the Ethics Committee in Research from the Federal University of Amapá (UNIFAP) (CAAE 42064815.5.0000.0003, Permit number 1.013.843).

## MATERIAL AND METHODS

### Study area

The study was conducted in the Araguari River basin, upriver of the newly installed Cachoeira Caldeirão Dam, in the state of Amapá, Brazil (N 0.77327, W 51.58064; Fig. 1). The regional climate is classified by Köppen-Geiger as Am (Equatorial monsoon) (*Kottek et al., 2006*), with an annual rainfall greater than 2,000 mm (*ANA, 2016*). The driest months are September to November (total monthly rainfall <150 mm) and the wettest months (total monthly rainfall >300 mm) from February to April (S1 Fig in *Paredes et al. (2017)*). The Araguari River rises in the Guianan Shield at the base of the Tumucumaque uplands and until recently discharged directly into the Atlantic Ocean (i.e., was not part of the Amazon River basin). However, in 2015 the river course changed due to as yet undefined anthropogenic effects and the Araguari River now discharges directly into both the Atlantic Ocean (120 km east from the Cachoeira Caldeirão Dam) and the Amazon River (100 km south from the Cachoeira Caldeirão Dam).

The Cachoeira Caldeirão is a large run-of-river dam (219 MW, dam height 20.6 m), with three turbines that became fully operational in January 2017 (*EDP Brasil, 2017*). The Cachoeira Caldeirão is the third and most recent addition to a series of dams along a 20 km stretch of the Araguari River. The reservoir was filled in January 2016, and the testing of the first installed turbine started on 24 February 2016 (*ANEEL, 2016*). The

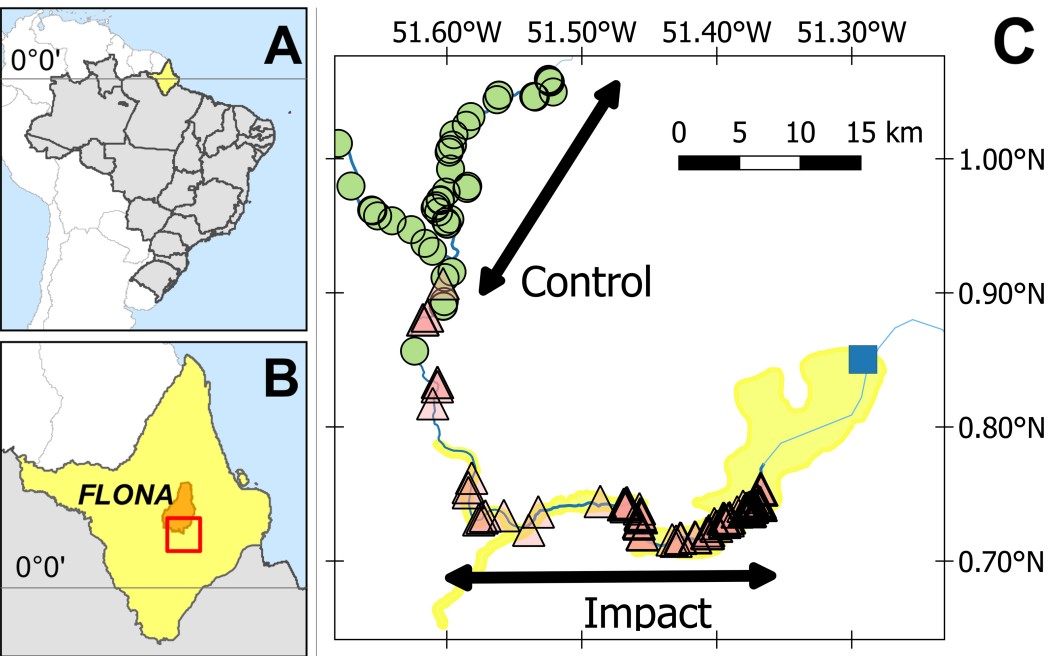

**Figure 1** **Study area.** (A) State of Amapá in Brazil. (B) Location within Amapá. (C) Showing location of submerged (triangles) and unsubmerged (circles) *Podocnemis unifilis* nest-areas. Yellow shading delimits the area directly impacted by the Cachoeira Caldeirão Dam (blue square) as defined by the environmental impact assessment. Location of the multiyear (2011, 2015 and 2016) control-impact survey areas are indicated along the rivers. Impact surveys were conducted along the directly impacted river section and control surveys along a river section between 22.9 and 55.9 km upstream of the directly impacted zone.

approved Cachoeira Caldeirão reservoir extended to an altitude of 58.3 masl, with a volume of 230.56 hm$^3$, covering an area of 47.99 km$^2$ (including 24 km$^2$ of flooding beyond the original river channels) and an average depth of 4.8 m (*EDP Brasil, 2017*).

The Cachoeira Caldeirão dam and hydropower plant was built, installed and is currently operated by the Empresa de Energia Cachoeira Caldeirão S.A, which is a joint venture between China Three Gorges Brasil (member of the China Three Gorges Corporation) and EDP—Energias do Brasil S.A (subsidiary of EDP—Energias de Portugal) (*Bloomberg, 2017*). Environmental Impact Assessments (EIAs) are required before any of the installation work can be started (*Fearnside, 2016*). These EIAs are paid for by the operating company (Empresa de Energia Cachoeira Caldeirão S.A), who contract companies (in this case Eletronorte, Odebrecht and Neoenergia) to evaluate the overall viability of the project. These companies subsequently subcontract the EIA to specialized consultancy firms.

## Study species

The semi-aquatic yellow spotted river turtle (*P. unifilis*) is widely distributed across lentic and lotic waterways in the Amazon and Orinoco river basins (Brazil, Venezuela, Columbia, Ecuador, Peru, Bolivia, Guyana, French Guiana and Suriname) (*Vogt, 2008*). Although *P. unifilis* is widespread, anthropogenic impacts such as hunting, nest harvesting, and deforestation means that the species is classified as Vulnerable (A1acd) by the IUCN

(*Tortoise & Freshwater Turtle Specialist Group, 1996*; *Vogt, 2008*). As found in many turtle species, nest site selection is an important component of *P. unfilis* demographics (*Escalona, Valenzuela & Adams, 2009*; *Iverson, 1991*). Females are thought to lay eggs once a year, with the timing of nesting synchronized with seasonal periods of low water levels (*Ojasti, 1996*; *Vogt, 2008*). Females can lay nests in a wide variety of substrates (*Escalona, Valenzuela & Adams, 2009*; *Foote, 1978*; *Pignati et al., 2013*), with nesting recorded in pasture (*Ramon dos Santos, 2013*) and even on top of caiman nests (*Maffei I & Da Silveira II, 2013*). However, such examples are atypical. More typically *P. unifilis* nesting shows a nonrandom pattern, with nesting areas sharing similar environmental characteristics such as beaches and river banks with dry and relatively exposed sandy-silty substrates (*Escalona, Valenzuela & Adams, 2009*; *Pignati et al., 2013*).

## Nest-area surveys

Surveys were conducted between September and December in three nesting seasons, two pre- (2011, 2015) and one post- (2016) reservoir formation. These months correspond to low water and include the complete nesting and first half of the hatching season in the study area (D Norris pers. obs., 2016). Nesting area data from 2011 were obtained from a previous study along 66 km of river (*Arraes, 2012*). In 2015 and 2016, we then repeated and extended the methodologies applied in 2011, along 150 km of river upstream of the Cachoeira Caldeirão dam (Fig. 1).

Surveys started 12 km upstream of the dam. This start point was established in 2011, and in 2015/2016 it was not possible to safely conduct monthly surveys closer to the dam due to the operations that coincided with the survey period (e.g., involving boats, dredgers, and construction). Surveys were used to identify and map nesting areas. Originally in 2011, two separate 33 km river sections were surveyed for nesting areas, the first starting 12 km upstream from the dam. The 2011 survey methodology was extended in 2015 and 2016, when we surveyed a continuous 150 km stretch of river and also considered both potential and actual nesting areas. Potential nesting areas are all locations that had suitable habitat conditions for nesting but where no nests were found and actual nesting areas those locations where females actually nested.

Methods used to identify turtle nest-areas were standardized among years. To minimize possible detectability bias related to the searches of turtle nesting areas and nests we maintained at least one observer in the team constant while conducting searches in all years (2011, 2015, and 2016). Monthly boat surveys were used to identify nesting areas. While navigating along the rivers in a motorized boat at a constant speed (ca. 10 km/h) we performed an extensive search for nesting areas that involved identifying potentially suitable areas through visually searching river banks, circling islands, stopping to search among boulders and rapids. These searches were conducted together with local residents with over 30 years of knowledge of nesting areas. We identified potential areas (Figs. S1A–S1B) where environmental conditions matched those described in the literature (*Escalona, Valenzuela & Adams, 2009*; *Pignati et al., 2013*) and/or those found at the nesting areas from 2011. These potential areas were identified based on the presence of at least 1 m$^2$ of exposed sand and/or fine gravel substrate, raised sufficiently high above the river level to not be

waterlogged at a depth of 15 cm, which is representative of the depth females dig when nesting (depth to first egg is typically 7–10 cm,  D Norris pers. obs., 2016 (*Pignati et al., 2013*)). Based on the diversity of nesting areas along 150 km of river in the study area (Fig. S1), we did not include slope as a selection criteria. Sites that remained waterlogged (on or close to the water level) or predominantly covered with an inappropriate substrate type (e.g., clay, large pebbles) were characterized as not suitable for nesting (Figs. S1D–S1E).

Potential areas were repeatedly surveyed to locate nests (each area being surveyed three times per season with an interval of 20–30 days between visits), once a nest was confirmed the area was considered as an actual nesting area. All nesting areas were characterized in terms of size and shape by mapping in the field using a handheld GPS and confirmed against high-resolution satellite images (RapidEye, 5 m resolution, downloaded from http://geocatalogo.mma.gov.br; Tile IDs: 2239415 (date 10/09/2015), 2239412 (date 22/09/2015) and 2239413 (date 20/10/2015)). We mapped only the surface area considered to be suitable for turtle nesting, i.e., excluding areas covered by rock/prominent boulders/roots/clay substrates.

To locate river turtle nests we conducted monthly surveys of all nesting areas. Nests were located by following turtle tracks on the sandy/gravel substrates and systematic substrate searches. Searches were conducted by a team of three observers at a standardized speed (mean 0.8, range 0.2–1.3 km per hour) and the time spent searching sites ranged from 10 to 97 min depending on the size of the site. To identify actual nest-areas, we considered the presence of all nests including those predated by humans or wildlife. Depredated nests were identified by the presence of broken eggshells outside the nest, disturbed/uncovered nests and the presence of excavation marks.

Female turtles are thought to repeatedly use the same nesting areas, but considering the possibility that there could be differences in the locations of nest-areas used between years and that despite our intensive effort we may have failed to detect some nests, we also conducted interviews with local residents to identify actual river turtle nest-areas during the last 5 years (*Norris & Michalski, 2013*). Thus, nest areas that were reported by interviewees and those where nests were identified during field surveys were both included as actual areas in our study. Sites with river turtle tracks but no confirmed nesting were included in the potential nesting area count.

## Data analysis

All statistical analyses were undertaken within the R language and environment for statistical computing (*R Core Team, 2017*). Variation in the number of nesting areas per km was examined using a generalized linear model (GLM) with Tweedie error distribution family (*Dunn, 2017*). Tweedie is a probability distribution family that includes the purely continuous normal and gamma distributions, the purely discrete Poisson distribution, and the class of compound Poisson–gamma distributions (*Jorgensen, 1997*; *Tweedie, 1984*). A maximum likelihood method (function "tweedie.profile") was used to profile the Tweedie variance function index parameter $p$. The profiling generated an estimate of 1.155, which corresponds to a compound Poisson–gamma distribution (*Jorgensen, 1997*). This compound Poisson distribution is particularly useful for modelling responses with a

continuous distribution on positive values but with some observations being exactly zero (*Jorgensen, 1997*). The error distribution family was modelled with the tweedie package default log-link power function.

To represent the BACI design, the response of actual nesting beaches per km was modelled against two fixed factors, (i) before-after and (ii) control-impact. The impacted area corresponds to 33 km of the Araguari River, within the area of direct impact (as defined by the EIA (*Ecotumucumaque, 2013*; *EDP Brasil, 2017*), 12–45 km upstream of the dam (Fig. 1). For control areas, we used a 33 km stretch of the Falsino river 61–94 km upstream of the dam (Fig. 1), which corresponds to a region with little anthropogenic disturbance, with less than five households (*Norris & Michalski, 2013*) and fisherman are not allowed to enter. For the GLM analysis each area (control, impact) was divided into five subsections with an equal length of 6.6 km (Table S1). This distance was chosen to provide a representative number of spatially independent replicates for model estimation. Greater numbers of shorter sections resulted in higher proportions of river sections with zero nesting areas, which consequently reduced the performance of GLM estimation and fit. We also included the interaction between factors, where a significant interaction represents differences between the impacted area compared with the control after the reservoir filling (*Underwood, 1993*).

To examine the spatial extent of submersion we considered potential and actual nest-areas up to the farthest submerged nest-area. To determine the distance of each nest-area from the dam, we obtained the center of each nest-area polygon and then calculated the distance along the river between each center point and the dam using functions available in the R (*R Core Team, 2017*) package *riverdist* (*Tyers, 2017*). As continuous surveys of nesting areas were not conducted in 2011, analysis of the spatial extent of nest-area submersion was conducted by comparing data collected in 2015 (pre-reservoir formation) and 2016 (post-reservoir formation). A nesting area could be either potential or actual during the nesting period. That is, if a potential area was identified and in a subsequent visit a nest was found then the area was counted only as actual during the nesting season.

## RESULTS

Prior to reservoir filling, we encountered on average 0.5 river-turtle nesting areas per km along 66 km of rivers (Fig. 2). Although there tended to be more nesting areas in control sections (0.7 per km compared with 0.4 per km in the impacted sections), there was no significant difference between the mean numbers of nest-areas encountered in control and impacted river sections prior to reservoir filling (Table 1, Table S1, Fig. 2 (95% confidence intervals overlap sample means)). Following reservoir filling there was a drastic reduction in the number of nesting areas encountered along the 33 km within the impacted area (0.065 areas per km, Fig. 2). This decline was a clearly different pattern from the relatively stable number of nesting areas encountered in the control sections (Fig. 2, Table 1 GLM interaction term $P = 0.041$).

Reservoir filling reduced the overall (potential and actual) nesting area to 17% of the pre-filling baseline along 57.4 km of river upstream of the dam (Table 2). A total of 3.8 ha

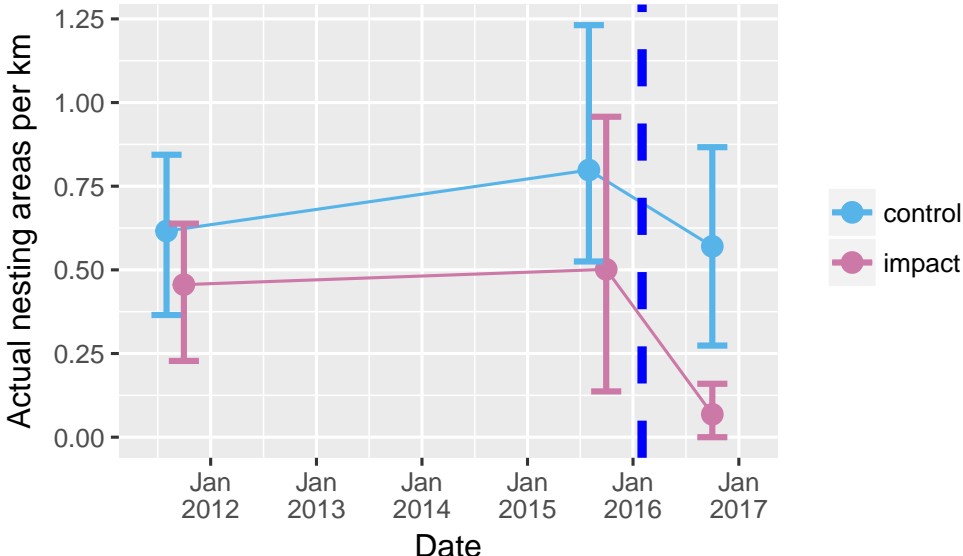

**Figure 2   Nesting areas before and after reservoir filling.** Number of yellow spotted river turtle nesting areas encountered during two nesting seasons before (2011, 2015) and one nesting season following (2016) reservoir filling of the Cachoeira Caldeirão Dam. Nesting area surveys were conducted along 33 km in both control (61–94 km upstream) and directly impacted (12–45 km upstream) river sections. Points show means and solid vertical lines are 95% confidence limits estimated via nonparametric bootstrap. Surveys were conducted simultaneously but points have been dodged along the $x$-axis for clarity. Dashed vertical line represents when the reservoir was filled.

**Table 1   Comparison of nesting areas in control and impacted river sections before and after reservoir filling.** Results from GLM used to explain the variation in the number of nesting areas per km of river, recorded before (2011, 2015) and after (2016) hydropower reservoir filling. Nesting area surveys were conducted along 33 km in both control (61–94 km upstream) and directly impacted (12–45 km upstream) river sections in all years.

| Source of variation | Actual nesting areas per kilometer | | | |
|---|---|---|---|---|
| | Estimate | SE | T value | P[a] |
| (Intercept) | 1.08 | 0.05 | 21.90 | <0.001 |
| Before-after (compared with 2011) | | | | |
|     2015 | −0.04 | 0.06 | −0.66 | 0.519 |
|     2016 | 0.01 | 0.07 | 0.18 | 0.857 |
| Control-impact (impact vs control) | 0.05 | 0.08 | 0.68 | 0.506 |
| Interaction (Before-after: control-impact) | | | | |
|     2015: impact | 0.02 | 0.10 | 0.25 | 0.804 |
|     2016: impact | 0.37 | 0.14 | 1.89 | 0.041 |
| Observations | 30 | | | |
| Model deviance explained (%) | 37.0 | | | |
| Model P[b] | 0.0398 | | | |

**Notes.**
[a]Factor $P$ values obtained from comparison against the $t$ statistic probability distribution.
[b]Model $P$ value obtained from comparison against single factor (control-impact) null model.

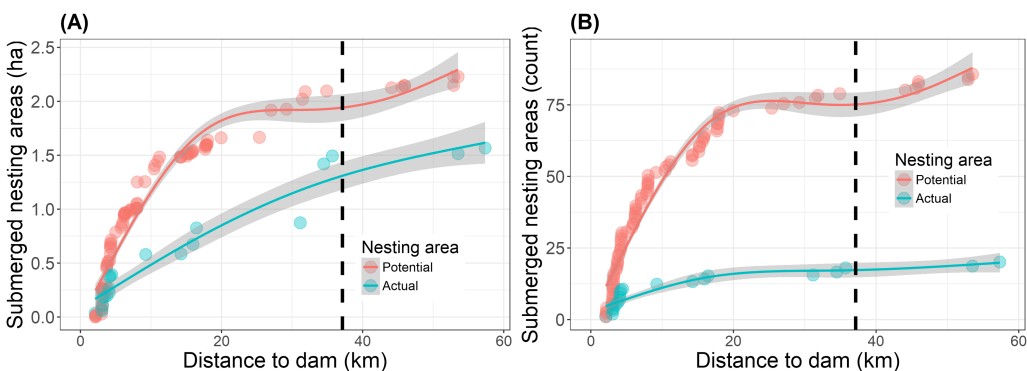

**Figure 3  Submersion of river turtle nest-areas.** (A) Cumulative area (ha) and (B) cumulative count of nest-areas submerged by reservoir formation at the Cachoeira Caldeirão Dam, Amapá, Brazil. Potential nest-areas had suitable habitat for nesting but no nests were detected and actual are those areas where females nested in 2015 and/or in the previous five years (2010–2014). Dashed vertical line represents the limit of direct impact defined by the environmental impact assessment. Lines and shaded areas are mean values and 95% confidence intervals from Generalized Additive Models (formula $= y \sim s(x, k = 4, bs = ``cs")$) that are added as a visual aid to illustrate trends in the cumulative values.

**Table 2  Nesting areas encountered along the Araguari river basin.** Comparison of nesting areas recorded before (2015) and after (2016) hydropower reservoir filling. Potential nest areas had suitable habitat for nesting but no nests were detected and actual areas are where females nested in 2015 and/or in the previous five years (2010–2014).

| Year | No. nesting areas (potential, actual) | Total nesting area (ha) (potential, actual) | Mean nesting area (ha) (potential, actual) | Nest density[a] (N) | Human removal (N) |
|------|---------------------------------------|---------------------------------------------|---------------------------------------------|---------------------|-------------------|
| 2015 | 114 (80, 34) | 5.2 (2.61, 2.60) | 0.03 (0.03, 0.08) | 18.9 (49) | 76% (37) |
| 2016 | 29 (15, 14) | 0.9 (0.38, 0.56) | 0.04 (0.03, 0.04) | 53.6 (30) | 77% (23) |

**Notes.**
[a] Nests per ha of actual nesting areas.

of nest-areas (both potential and actual) were submerged as a direct result of the reservoir filling in 2016 (Fig. 3, Table S2). Of this total, 1.6 ha of actual nest-areas were submerged and 13% (0.2 ha) of these actual nest-area losses occurred beyond the legally defined limit of direct impact (Fig. 3). The farthest submerged nest-area was a 520 m$^2$ actual nest-area, located 57.4 km from the dam and 20.2 km upstream of the limit of direct impact defined by the environmental impact assessment (Fig. 3). Beyond this point none of the areas recorded in 2015 were submerged. Repeating the same nest surveys as in 2015, we also found three new actual nest-areas in 2016.

In addition to complete submersion, there was a drastic reduction in the size of the remaining actual nesting areas (Table 2). Overall fewer nests were encountered in 2016, but the density of nests at the remaining sites increased nearly three-fold. This increased density was correlated with both a reduction in nest-area area and an increased number of nests per area, with the mean number of nests encountered increasing by 50% from the 2015 baseline (from 1.4 to 2.1 nests per area in 2016). Overall nest mortality was high, with only one nest successfully hatching in 2015 and none in 2016. In both years the majority

of nests were removed by humans (Table 2). Areas with multiple nests were most strongly affected by humans (mean removal 98%), whereas there was no human removal of nests from areas with only one nest.

## DISCUSSION

For the first time we present a representative and robust before-after control-impact comparison that establishes a minimum value for the spatial extent of the direct impacts of a hydropower development on an Amazon river turtle. Our findings support a growing body of research that shows that without direct conservation intervention river turtles are unlikely to survive the severity and speed of environmental changes caused by hydroelectric development (*Ihlow et al., 2012*; *Rhodin et al., 2011*; *Rödder & Ihlow, 2013*). We first explore the extent of nest-area lost due to submersion, and then turn to explore how negative impacts caused by a dam reservoir can extend beyond environmental impact assessment limits. Finally, we discuss ways to mitigate the negative effects of dam construction in freshwater river turtles.

### Extent of nest-area loss/submersion

Our results showed that a total of 3.8 ha of nest-areas were submerged as a direct result of the new hydropower reservoir filling in 2016, which accounted for the loss of 85 (25.4%) actual/potential nest-areas in the study area. We cannot attribute nest-area differences across years to detectability bias of observers as we maintained at least one observer from the local community constant in all three survey years (pre- and post-dam). Thus, we can assume that differences reported here between year comparisons are not related to any systematic bias in methods and/or observers.

Nest-areas are representative of river turtle populations and have been studied for several purposes but mainly for population estimation (*Pignati et al., 2013*) and evaluation of sustainable harvest (*Caputo, Canestrelli & Boitani, 2005*). Yet traditional population demographic parameters are time consuming to obtain at the scale of areas affected by hydropower developments and will not necessarily provide information that is urgently needed to inform effective conservation actions and solutions. We, therefore, present representative data on nest-area submersion that reflects negative effects caused by the new hydropower dam construction on freshwater turtles.

We found that the dam of a hydropower development permanently submerged actual and potential nest-areas where *P. unifilis* used to make nests. Similar results have been found across the Amazon, with studies showing that hydropower disruption of flows and natural seasonal flood-pulses severely compromises biodiversity along Amazon rivers (*Castello et al., 2013*; *Fearnside, 2009*; *Latrubesse et al., 2017*; *Lees et al., 2016*; *Sousa Júnior et al., 2016*; *Winemiller et al., 2016*). Thus, with a quarter of the actual/potential nest-areas used by turtles in our study area directly affected by water submersion, we showed that females lost areas used for nesting and will need to find new nesting sites. As female turtles are thought to exhibit a strong degree of nest-area fidelity (*Valenzuela & Janzen, 2001*) we anticipate increased adult mortality as a direct result of loss of nest areas. Females will have to search for new areas, which increases exposure to predators and energy expenditure

at a time when females need to invest in development of eggs. Additionally, dispersal movements used to fulfill reproductive life requirements due to the new reservoir filling may generate conflicts with other resident turtles (*Alho, 2011*), rendering energy demand and turtle density alterations with potential conflict for food resource and non-flooded potential nest-areas.

Environmental changes caused by hydropower development will not only reduce nesting success but will also cause outright loss of population segments. For example, in North America, approximately 6–98 m of land is required to encompass each consecutive 10% segment of a freshwater turtle nesting population up to 90% coverage, with ca. 424 m being required to encompass the remaining 10% (*Steen et al., 2012*). While some freshwater turtles require modest terrestrial areas (<200 m zones) for 95% nest coverage, others require larger zones (*Steen et al., 2012*). Additionally, a 30 year study of Blanding's Turtles indicated that 39% of females and 50% of males captured, maintained the same residence wetland for over 20 years in southeastern Michigan (*Congdon, Kinney & Nagle, 2011*).

We expect that loss of potential nesting areas will concentrate and intensify human nest removal on remaining areas. We found that the density of nests at the remaining un-flooded areas increased by nearly three-fold with an increased number of nests per area. Changes in turtle nest densities have been shown to increase predation, with clumped nests depredated at a greater rate than scattered nests (*Marchand et al., 2002*). Considering that *P. unifilis* is widely consumed and traded in the Amazon region (*Peres, 2000*; *Pezzuti et al., 2010*; *Smith, 1979*), and eggs and adults are also being consumed in our study region (*Norris & Michalski, 2013*), we anticipate that human nest removal is likely to increase. Within our study area humans are by far the dominant predator of turtle nests. Currently, the local riverine people remove 76–77% of turtle nests. Therefore, the reductions in areas where nests can be laid coupled with increased nest density in a smaller number of areas will likely concentrate and intensify human nest removal on the few remaining areas. The consumption of river turtle eggs by local people may also increase due to the complex social impacts associated with dam and reservoir creation, such as the resettlement of displaced populations, loss of fish and other resources by riverine communities (*Fearnside, 2001*; *Tilt, Braun & He, 2009*), coupled with the destruction of the previously occupied physical space (*Finley-Brook & Thomas, 2010*).

## Implications for restoration of freshwater ecosystems

Our findings support the view that when placed within scenarios of rapid and poorly planned anthropogenic development, the recovery of biodiversity (from ecosystems to populations) is unlikely without direct interventions (*Latrubesse et al., 2017*; *Lees et al., 2016*; *Tundisi et al., 2014*). BACI is not required for Brazilian environmental impact assessments. This means that environmental impact assessments generally lack scientific rigor, depending largely on simulations to predict impacts and space-for-time comparisons for evaluation and monitoring. Remote sensing is a step forward and has been used to generate detailed understanding of broad regional scale impacts and consequences of hydropower developments in the Amazon basin (*Fearnside, 2009*; *Fearnside & Pueyo, 2012*; *Latrubesse et al., 2017*; *Stickler et al., 2013*). Yet the lack of BACI studies at the local scale

limits our ability to generate biodiversity compensation or restoration actions. Recent reviews demonstrate a lack of robust data (*Alho, 2011*; *Lees et al., 2016*). Such data is necessary to inform effective impact assessments, and its absence limits the efficiency of proposed management procedures for conservation, restoration and sustainable use.

Our results show that restoration actions for our study area must replace at least 1.5 ha of suitable substrate for actual river turtle nest areas. If we consider the total area (actual + potential), then at least 3.8 ha of suitable nesting habitat must be restored. The enhancement/formation of reservoir islands for conservation is also necessary and has already been proposed to mitigate negative effects of dams (*McCartney, 2009*). Such actions enable the integration of social (as points for leisure and environmental education) and biodiversity conservation objectives. Yet such restoration actions are unlikely to succeed in isolation. Considering the levels of human nest removal, additional measures will be necessary for the long term conservation of river turtles in the area. Environmental education campaigns aiming to reduce turtle egg consumption, a cultural habit established in the study area (*Norris & Michalski, 2013*), will be a critical component for the timely mitigation of the negative impacts of dam constructions.

Humans are compensated (albeit often unfairly/incompletely) when their land has been completely or partially submerged by a hydropower reservoir (*Fearnside, 2001*; *Fearnside, 2016*; *Tilt, Braun & He, 2009*). Yet restoration actions have not been anticipated and/or implemented for this widespread semi-aquatic species of turtle, which holds cultural, economic and ecological significance. Brazil has the legal structure to implement statutes and legally oblige developments to implement biodiversity/environmental compensation actions. A recent example of Brazilian legislation generating positive biodiversity outcomes is the case where anticipated hydropower development impacts on indigenous lands halted (at least temporarily) one hydropower development in Mato Grosso State (*Bergen, 2016*). But the impacts of other neighboring dams in this same area show the harsh reality. Whilst other nations strengthen environmental laws, Brazilian legislation allows sacred areas to be dynamited and provides awards to those responsible (*Branford & Torres, 2017*). Unfortunately, recent proposals, under consideration in the Brazilian congress ("Câmara dos Deputados") are likely to further weaken existing environmental legislation (*Azevedo-Santos et al., 2017*; *Fearnside, 2016*). These cases highlight the continued need for robust science, including data collected from BACI designs to inform ongoing political debates within Brazil.

## CONCLUSIONS

We conclude that freshwater turtles considered in this study were highly vulnerable to nest-area losses due to submersion of actual and potential nest-areas in 2016 as a direct result of river level rises caused by the Cachoeira Caldeirão Dam. The synergistic effects of the construction of the new reservoir coupled with the alterations in nest density in the remaining nest-areas will likely affect negatively freshwater turtles. Additionally, the social impacts on human riverine populations and the loss of resources such as fish are likely to increase the consumption of freshwater turtle eggs. These factors are

likely to act synergistically with nest area losses to drastically reduce freshwater turtle populations. Despite the at least temporary persistence of *P. unifilis* nests in the study area, there is little evidence that this species will be able to cope with the severe reduction in nest-areas associated with human removal of nests in the near future. Populations of freshwater turtles are likely to succumb if mitigation is not undertaken both during and after reservoir formation. However, additional research on post-dam construction effects such as population level information on how freshwater turtles deal with such drastic environmental changes are necessary to support the development of effective conservation solutions. As a minimum conservation strategy, mitigation of pervasive negative effects of the new reservoir should be implemented to ameliorate devastating effects on turtles and other semi-aquatic species.

## ACKNOWLEDGEMENTS

The Instituto Chico Mendes de Conservação da Biodiversidade (ICMBio) and the Federal University of Amapá (UNIFAP) provided logistical support. We are grateful to Alvino Pantoja Leal, Cremilson, Cleonaldo and Cledinaldo Alves Marques for their invaluable assistance during fieldwork. We thank the editor Jonathan D. Tonkin and Erin Abernethy and Daniel Escoriza for their helpful revision of earlier versions of the text.

### Funding

Funding was provided by the National Academy of Sciences and the United States Agency for International Development through the Partnership for Enhanced Engagement in Research (http://sites.nationalacademies.org/pga/peer/index.htm), and award number AID-OAA-A11-00012 to Darren Norris, James P. Gibbs and Fernanda Michalski. The funders had no role in study design, data collection and analysis, decision to publish, or preparation of the manuscript.

### Grant Disclosures

The following grant information was disclosed by the authors:
National Academy of Sciences.
United States Agency for International Development: AID-OAA-A11-00012.

### Competing Interests

The authors declare there are no competing interests.

### Author Contributions

- Darren Norris conceived and designed the experiments, performed the experiments, analyzed the data, contributed reagents/materials/analysis tools, wrote the paper, prepared figures and/or tables, reviewed drafts of the paper.
- Fernanda Michalski conceived and designed the experiments, performed the experiments, contributed reagents/materials/analysis tools, wrote the paper, reviewed drafts of the paper.

- James P. Gibbs conceived and designed the experiments, wrote the paper, reviewed drafts of the paper.

## Field Study Permissions

The following information was supplied relating to field study approvals (i.e., approving body and any reference numbers):

Permission to collect observational data from river turtle nest-areas was provided by research permit number IBAMA/SISBIO 49632-1 and 49632-2 to DN and FM, issued by the Instituto Chico Mendes de Conservação da Biodiversidade (ICMBio). Interviews with local residents were approved by IBAMA/SISBIO (permits 45034-1, 45034-2, 45034-3) and the Ethics Committee in Research from the Federal University of Amapá (UNIFAP) (CAAE 42064815.5.0000.0003, Permit number 1.013.843).

## Data Availability

The raw data is provided as Supplemental Files.

## Supplemental Information

Supplemental information for this article can be found online at http://dx.doi.org/10.7717/peerj.4228#supplemental-information.

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
