# Peer review of "Beyond harm’s reach? Submersion of river turtle nesting areas and implications for restoration actions after Amazon hydropower development"

_PeerJ, doi:10.7717/peerj.4228_

## Round 0.1 · original submission · Minor Revisions

· Academic Editor

Minor Revisions

Your manuscript attracted considerable interest from the two reviewers. Both reviewers highlight important issues, and offer a number of suggestions for improvement. For instance, Reviewer 2 highlights that more details are needed on the GLM methodology and reporting. In that light, more details are required on the results of the statistical tests in general. Rather than say ‘GLM interaction term P < 0.05’ (L 170), and ‘confidence intervals overlap means’ (L 167), please give the actual values and the results of the full GLM. I agree with Reviewer 1 that more detail is needed regarding the methods for determining what ‘suitable habitat conditions for nesting’ is. There is a consistent, quantifiable method behind this I assume? I’m also inclined to agree with Reviewer 2 that the discussion should be kept as on-topic as possible with regard to the science (e.g. L268-279), and I would caution that any extra speculation on the human element of dam effects as highlighted by Reviewer 1 should only be included if deemed absolutely necessary (and, if so, be kept very brief). Finally, the raw data isn’t sufficient to reproduce the results. Could you please upload the more detailed raw data that was used in your analyses?

On the balance of these two reviews, I invite you to resubmit a revised version of your manuscript in light of their comments. I look forward to seeing a revised version. Please see below some more minor comments from me.

Minor comments:

Details of survey timing. Could you please be more specific about whether the timing of surveys captured both nesting and hatching and if there was any issues with spatial variability or bias in the timing of observations across the study region.

Details of the dam operation. Is it possible to provide details on how the reservoir will fluctuate under future management (e.g. will there be late summer draw-downs etc.?) and how this will further affect turtle populations?

‘River’ should be capitalized in ‘Amazon River’

L22: Comma at end of line should be semi colon.

L105: I find this statement a little confusing. Could you be more clear or just state it was diverted to join the Amazon basin without giving a reason?

L171-174: I can’t figure out where these numbers are coming from specifically; e.g. the 3.8 and 1.6 ha. Could you double check they’re correct please.

L282: ‘losses’.

L290: Better to state ‘such as’ or another alternative than placing ‘e.g.’ mid-sentence.

References: Please ensure references are complete. e.g. L393.

Fig. 1: Caption should reflect that the yellow section is also the ‘impact’ section for the purposes of the analyses (if I’m judging this correctly). It’s not clear to me exactly where the boundaries of the control and impact sections are on the map by looking at the ‘control’ and ‘impact’ text and arrow bars, unless the yellow section is more clearly linked with the text that says ‘Impact’ on the map.

Table 1: Be clear why the 2011 data wasn’t used here.

Fig. 3: More details are needed for the methods behind this figure in the methods section. e.g. count vs. cumulative, LOWESS smoothing line.

·

Basic reporting

Overall, the writing and style of the manuscript is high quality, concise, and easy to follow. The paper utilizes pre- and post-dam data to analyze the impact of a dam’s reservoir on one species, which is an important, but many times neglected, element of understanding the realized impact of hydropower projects. A minimal amount of text (specific suggestions given) added to the methods and results will aid in reader understanding.

Given the high predation of turtle nests by humans, the authors should acknowledge the complex impact of dams on local humans, specifically that human-ecosystem relationships change in ecosystems altered by dams. Human predation on the turtles is discussed in one paragraph of the discussion, and there is just one sentence stating that humans in the submergence zone are compensated but that these turtles have not been considered. No acknowledgment is made by the authors that this compensation to humans is rarely, if ever, equivalent to the economic loss suffered by the displaced peoples (McCully 2001). It would be appropriate to briefly discuss the situation of these displaced peoples, if known from personal communication (potentially 244 families according to the EDP annual report 2013), and how this could potentially relate to the increased pressure on turtle nests, especially given the higher density of (and ease of finding) nests. Given the impact of dams on turtle nest-areas, it seems like decreasing human predation on turtle nests would be a critical component of turtle conservation in this area and a necessary component of mitigation efforts (in addition to those mitigation efforts suggested by the authors: BACI studies, restoring bank and island habitat, and protective legislation).

Experimental design

The experimental design is well suited to answer the proposed research question, how far upstream the reservoir impacted one species of turtle and whether this result was accurately reflected in the environmental impact assessment.

Line 123: Remove “potential and effective” before turtle nest-areas, as the description comes in the following paragraph.

Line 130: Please add one sentence clarifying how you determined potential nest-areas. Was this done by walking the banks and mapping all areas in the 66 km stretches of riverbank not covered by rock/boulders/root/clay substrates? Please move the other sentence describing potential nest-areas (Lines 146-147) to this earlier description. Please also state in this description that potential does not include effective (this was made clear by Table 1).

What about the nest-area designations being potential and actual (instead of effective) nest-areas? Effective may suggest to some readers that the nest was successful in producing young.

Validity of the findings

I found the majority of the results presented in a clear manner. Readers would be interested in the authors’ speculation in the discussion on how changing water levels in the reservoir would affect turtle nesting habitat and how the dam and altered flows could affect migration patterns.

Line 171: Why do you state the percentage of area that is reduced along 60.1 km and not the full study area of 66 km?

Figure 3: Panels A and B are redundant. Please remove. “C” should be capitalized. Please report an equation and R^2 values on the plots for the lines of fit.

Comments for the author

Text to add to the “Study area” section: What type of dam is Cachoeira Caldeirão? Who paid for the dam and completed and paid for the EIA?

Be consistent in using pre and post vs pre- and post-

Include the product information for Rapideye

Clayey should be clay

Consistent spacing between numbers and units (Line 152)

Line 189: Remove the statement of novelty

Line 233: Mention that the Congdon paper is referring to the Blanding’s turtle and the location of the study area.

Line 289: Add “are” to reservoirs are built

·

Basic reporting

Summary Norris et al. investigated the effect of the construction of a dam on the reproductive success of an aquatic turtle in the Amazon region. The results indicated that the construction of dams had negative effects on the reproductive success of this species. Additionally, the authors proposed some strategies to mitigate these effects.

Overview The article is well written and easy to follow. The analyses are adjusted to theobjectives of the study and the results are adequately discussed. The study has a strong point, which is that it compares the success of nesting before and after the river regulation; however some sections of the manuscript would benefit from further literature support. The study has also important implications for the management of turtle populations in basins impacted by dams.

Experimental design

Overview The article is well written and easy to follow. The analyses are adjusted to the objectives of the study and the results are adequately discussed. The study has a strong point, which is that it compares the success of nesting before and after the river regulation; however some sections of the manuscript would benefit from further literature support. The study has also important implications for the management of turtle populations in basins impacted by dams.

Validity of the findings

The analyses are adjusted to the objectives of the study and the results are adequately discussed.

Comments for the author

Specific points
Abstract The introduction of the abstract is excessively long (5/13 lines). I suggest to the authors to reduce it. The authors could use this space to explain (in the last sentence) some of the mitigation proposals.
Introduction In general, it is well structured, although the authors could consider a more general approach at the beginning of the introduction (e.g. effect of dams on riverine biodiversity).
Ln 63-Ln 67. These effects have no interest in the current study, although they could be cited in an introduction to the ecological impact of dams/river regulation. Perhaps a general revision about these impacts could be placed at the beginning of the introduction (including population isolation, reservoirs for invasive species, …).
Ln 64-67. These lines could be moved to the end of the introduction and used to formulate and support a hypothesis.
Methods
The authors could briefly describe the climate of the study area. This would help the reader to understand the nest-area survey timings. They could also (briefly) discuss some aspects of the target species (eg distributional range, natural history -focusing in breeding aspects-, IUCN status) that authors believe could provide more background and strength to their discussion. Non-Brazilian readers may find the article interesting but largely ignore the geographical and ecological frame of this study system.
Ln 118. Are there no references in the literature describing the nesting season of P. unifilis?
Ln 121. I suppose that the difference in the length transects (2011 vs 2015/2016) is explained by the inclusion of a control transect. It should be detailed.
Ln 122. Authors have to specify which are these conditions and include appropriate references.
Ln 134. Authors could consider include references to this method of survey. What are the ‘systematic substrate searches’?.
Ln 158. The software used to implement the GLM should be specified. Also the distribution family assumed for the dependent variable (nesting beaches / km2).
Discussion
Ln 248. Authors should consider include these references.
Ln 268-279. I find this part of the discussion interesting and ‘ethically’ justified, but completely out of context. In my opinion, the discussion should focus on the subject of study. Here the authors could provide some examples (references) explaining that restoring nesting habitats in aquatic turtles is a successful strategy. This would support the mitigation measures proposed below.
Ln 281-294. Apart from the negative impact on nesting, the dam and river regulation may involve some other pernicious effects (some indirect, mediated by predators or competitors) on turtle populations??.
Authors should comment this here or just in the previous paragraph.

---

## Round 0.2 · Minor Revisions

· Academic Editor

Minor Revisions

Many thanks for your resubmission and your detailed response to the reviewers. I'm happy to see that the manuscript has improved substantially from the previous version. As a result, I decided not to send the manuscript out for another round of review. However, I have some remaining concerns that need attention, as outlined below in a revised version. Please also see the annotated pdf attached and be sure to use track-changes in the revised version.

1. GLM details: These still aren't specific enough. Tweedie error distribution family is broad. The following is taken directly from the reference you provided for the GLM "Produces a generalized linear model family object with any power variance function and any power link." You need to provide the exact error distribution and link functions. Best to provide the specific method used in R and the options chosen and what they mean statistically, and your reasons for this.

2. Determining potential nest-areas: What you have added certainly helps, particularly the photos. However, I still think you need to provide something more concrete. Was it only the local residents making the calls here as to whether sites were suitable or not, or did they pass that knowledge onto you so you could also judge? If so, what specifically were you looking for? Was it particular substrate types and sizes, bank slopes, shade etc. etc.? At the moment, you've stated what the technique was for looking, but not what you were looking for (although brief mention is made at L131-132 and L161). You don't have to provide all the minute details, but some idea of the criteria you were using would help in accompanying the excellent supplementary figure.

Table S1: Please specify that in the caption decimal places are represented by commas in the csv. Better yet, change them to decimals places if possible.

---

## Round 0.3 · accepted · Accept

· Academic Editor

Accept

Many thanks for improving your manuscript once again. I am delighted to accept the manuscript for publication. Please note in the proofing stage that there should be spaces between numbers and units in the newly-added text at L159-162 (e.g. 15 cm, not 15cm).